# Biotic assemblages of gelatinous zooplankton in the Gulf of Mexico and adjacent waters: An evolutionary biogeographic approach

**José María Ahuatzin-Hernández** [1]*, **Juan J. Morrone**[2], **Víctor M. Vidal-Martínez**[1]*

**1** Departamento de Recursos del Mar, Cinvestav Mérida, Mérida, Yucatán, México, **2** Departamento de Biología Evolutiva, Museo de Zoología 'Alfonso L. Herrera', Facultad de Ciencias, Universidad Nacional Autónoma de México (UNAM), México City, México

\* jose.ahuatzin95@outlook.com (JMAH); vvidal@cinvestav.mx (VMVM)

**Data Availability Statement:** All relevant data are within the manuscript and its Supporting information files.

## Abstract

Gelatinous zooplankton constitutes a polyphyletic group with a convergent evolutionary history and poorly known biogeographical patterns. In the Gulf of Mexico, a region with complex geological, hydrological, and biotic histories, the study of this group has been limited to taxonomical and ecological aspects. In this study, we implemented a track analysis to identify distributional patterns of gelatinous zooplankton in the Gulf of Mexico and adjacent waters based on a dataset of 6067 occurrence records corresponding to Hydrozoa, Scyphozoa, Cubozoa, Ctenophora, Chaetognatha, Thaliacea, and Appendicularia. Information was compiled from the Global Biodiversity Facility Information (GBIF) and Ocean Biodiversity Information System (OBIS) databases and peer-reviewed literature. Individual tracks were constructed by joining the minimum distance between the occurrence localities of each taxon using a minimum spanning tree algorithm. We identified generalized tracks using parsimony analysis of endemicity with progressive character elimination (PAE-PCE). The areas where different generalized tracks overlapped were considered to represent panbiogeographical nodes. Seven generalized tracks (two with nested patterns) and six panbiogeographical nodes were recognized, mainly in neritic zones. The distributional patterns of gelatinous zooplankton allowed us to identify four biogeographic areas, supporting previously proposed biogeographic schemes. Gelatinous zooplankton in the Gulf of Mexico showed a convergent spatial distribution that can be explained by vicariant and dispersal events. The historical biogeography of the gelatinous biotas of the Gulf of Mexico has been little studied compared to ecological approaches, and the lack of integrative studies considering historical patterns is evident. This type of research is fundamental to understanding the evolutionary history of natural resources from a spatial perspective, identifying sites of biodiversity and endemism, and establishing a biogeographic baseline of the region for further studies.

## Introduction

Gelatinous zooplankton, which includes cnidarians (medusae and siphonophores), tunicates (i.e., larvaceans and thaliaceans), chaetognaths, and ctenophores, constitutes a polyphyletic

**Funding:** J.M.A.-H. received a scholarship from Consejo Nacional de Humanidades, Ciencia y Tecnología (CONAHCyT) for his doctoral research (grant: 845170) The funders had no role in study design, data collection and analysis, decision to publish, or preparation of the manuscript.

**Competing interests:** The authors have declared that no competing interests exist.

group with diverse functional evolutionary convergences, such as soft, transparent bodies composed of more than 90% water, lack of exoskeleton, and the capability to reproduce quickly, forming massive aggregations (blooms) [1–4]. Some taxa in this group represent basal lineages of metazoans (i.e., ctenophores, medusae, and chaetognaths [5, 6]) with complex life cycles that are key to understanding the evolution and development of coloniality [7] or studying the interaction between benthos and the pelagic environments [8, 9]. Other taxa, such as thaliaceans and larvaceans, are active components of the zooplankton and, together with chaetognaths and ctenophores, play a crucial role in the dynamics of marine ecosystems [4, 10]. Hence, the study of gelatinous zooplankton as a functional group is promising to identify convergent evolutionary patterns and establish a biogeographic baseline regarding their spatial dynamics.

Gelatinous zooplankton plays a crucial role in biogeochemical cycles, and their members are primary consumers of phytoplankton and other zooplanktonic groups [2, 4, 11]. Therefore, food availability and other abiotic variables such as temperature, dissolved oxygen, and salinity have been identified as their main ecological drivers [2, 3, 12], whereas marine currents and phylogenetic traits such as life cycles act as historical drivers [13, 14]. The wide dispersal of some taxa due to marine currents and the apparent lack of physical barriers in the oceans [15] have traditionally been considered conceptual limitations for studying the distributional patterns of this group, so knowledge about its historical biogeography is poor. Nevertheless, the existence of a spatial structure in some members of the gelatinous zooplankton has recently been demonstrated, which is associated with the natural history of the taxa, highlighting aspects such as evolutionary radiation and ontogeny [14, 15], suggesting the possibility of conducting biogeographic studies on this group.

The Gulf of Mexico (GoMx) is a semi-closed oceanic basin that communicates with the Western Atlantic and the Mexican Caribbean [16, 17]. The geological history of the GoMx is long, dating to ~ 175–355 Ma ago, when Pangea began to rift [17]. Successive events of open-close, rotation, fracture, erosion, evaporation, submergence, and emersion have shaped this region [18]. The current oceanic circulation of the GoMx is dominated by the Caribbean Current System, which enters the Gulf through the Yucatán Channel and exits through the Florida Strait. On the eastern side, water flowing into the Gulf forms the Loop Current, a clockwise flow influencing the Northwest of the Yucatán Peninsula, Cuba, and the outer West Florida Shelf. The hydrological dynamism due to the Loop Current and its eddies play a key role in the distribution of diverse planktonic organisms [17, 19]. Therefore, spatial structuring in the GoMx's biotas, resulting from historical and oceanographic events, is expected.

Biogeographic studies with historical perspectives are scarce in the GoMx, and existing studies have focused on specific taxa [14, 20–24]. Other studies have divided the region into ecoregions and provinces based on hydrological, geological, and biological data, recognizing the Carolinian and Caribbean provinces [25, 26]. This scheme, however, does not necessarily reflect the distribution and endemism patterns of biodiversity in the region; therefore, it is necessary to establish a system of areas with these characteristics. The recognition of biotic assemblages (groups of taxa coexisting in space at a given time) is fundamental in biogeographical studies, as it is the first step in an evolutionary biogeographic scheme [27, 28] and is crucial for understanding the distribution of biodiversity, its correct management, and conservation [29, 30].

Track analysis, proposed by Leon Croizat, represents a valuable tool for analyzing historical distributional patterns [27, 31–33]. However, this method has been scarcely used in marine environments [e.g., 34–36] compared to terrestrial studies. Consequently, its relevance in these environments has not yet been explored. Track analysis allows us to propose primary biogeographical hypotheses (i.e., different taxa share congruent distributional patterns because

of a common biogeographic history) that can be tested through cladistic biogeographic analyses. Track analysis consists of three simple steps: 1) constructing individual tracks (connecting the localities of each taxon considering the minimum distance), 2) obtaining generalized tracks by the strict congruence of two or more individual tracks (if a distribution pattern repeats through different taxa, a biota can be recognized, and the space delimited by the coincident distributions becomes significant for a historical interpretation), and 3) identifying nodes in the areas where two or more generalized tracks intersect (the areas where two or more spatially restricted biotas occur may be characterized by high biodiversity, endemism, distributional boundaries, or disjunction and interpreted as biogeographical complex areas) [27]. Accordingly, this work aimed to implement a track analysis on gelatinous zooplankton of the GoMx and Western Caribbean to analyze its distributional patterns by identifying biotic assemblages, proposing hypotheses about the historical events that shaped its current distribution, and contributing to the establishment of a biogeographic baseline of the region.

## Material and methods

### Data acquisition and processing

We compiled data on the occurrence of gelatinous zooplankton in the GoMx and Western Caribbean, considering Hydrozoa, Scyphozoa, Cubozoa, Ctenophora, Chaetognatha, Thaliacea, and Appendicularia. Data were obtained by querying all available records for the previously mentioned taxa from the Global Biodiversity Facility Information (GBIF) [37] and Ocean Biodiversity Information System (OBIS) [38] databases, considering 'preserved specimens' and 'material citation' in the search. No other specific criteria were considered. In addition, a peer-reviewed literature search was conducted (S1 File) to supplement missing records from the digital databases. For Hydrozoa, we considered the literature list provided by Ahuatzin-Hernández et al. [14], excluding hydrozoans with a completely benthic life cycle and those exclusively developing medusoids or sporosacs. The information was merged into a final database, removing duplicate records and those falling in land. Taxon names were manually filtered through the World Register of Marine Species [39], correcting synonymies, invalid taxa, and typos. Taxa with only one record were omitted. The final database contains 6067 occurrences (Fig 1, S2 File).

### Track analysis

Track analysis is one of the most utilized methods for addressing historical biogeographic studies. Since its proposal [32, 33], various modifications and interpretations of the original ideas have been proposed [31]. Identifying biotic components as part of an integrative evolutionary scheme is one of the most recently implemented approaches [e.g., 40]. Here, we conducted a track analysis according to the proposal of Echeverry & Morrone [41], i.e., applying a parsimony analysis of endemicity with progressive character elimination (PAE-PCE). Individual tracks for each taxon were constructed in QGIS 3.32.2. using a minimum spanning tree (MST) algorithm with the MST package [42]. Individual tracks were then superimposed on a 2° x 2° grid of the study area and codified into a matrix of tracks per quadrants, coding 1 for presence and 0 for absence. A hypothetical area coded 0 was added to root the cladogram. Before analysis, we removed tracks occurring in only one quadrant (autapomorphies). The final matrix had 233 tracks (columns) and 70 quadrants (rows) (S3 File). This matrix was analyzed using a parsimony algorithm in TNT 1.6 [43], implementing Ratchet as the search strategy, random seed = 1, and find minimum length = 1. When more than one cladogram was retained, a strict consensus cladogram was applied. Clades supported by two or more synapomorphic individual tracks were treated as generalized tracks and interpreted as ancestral biotas

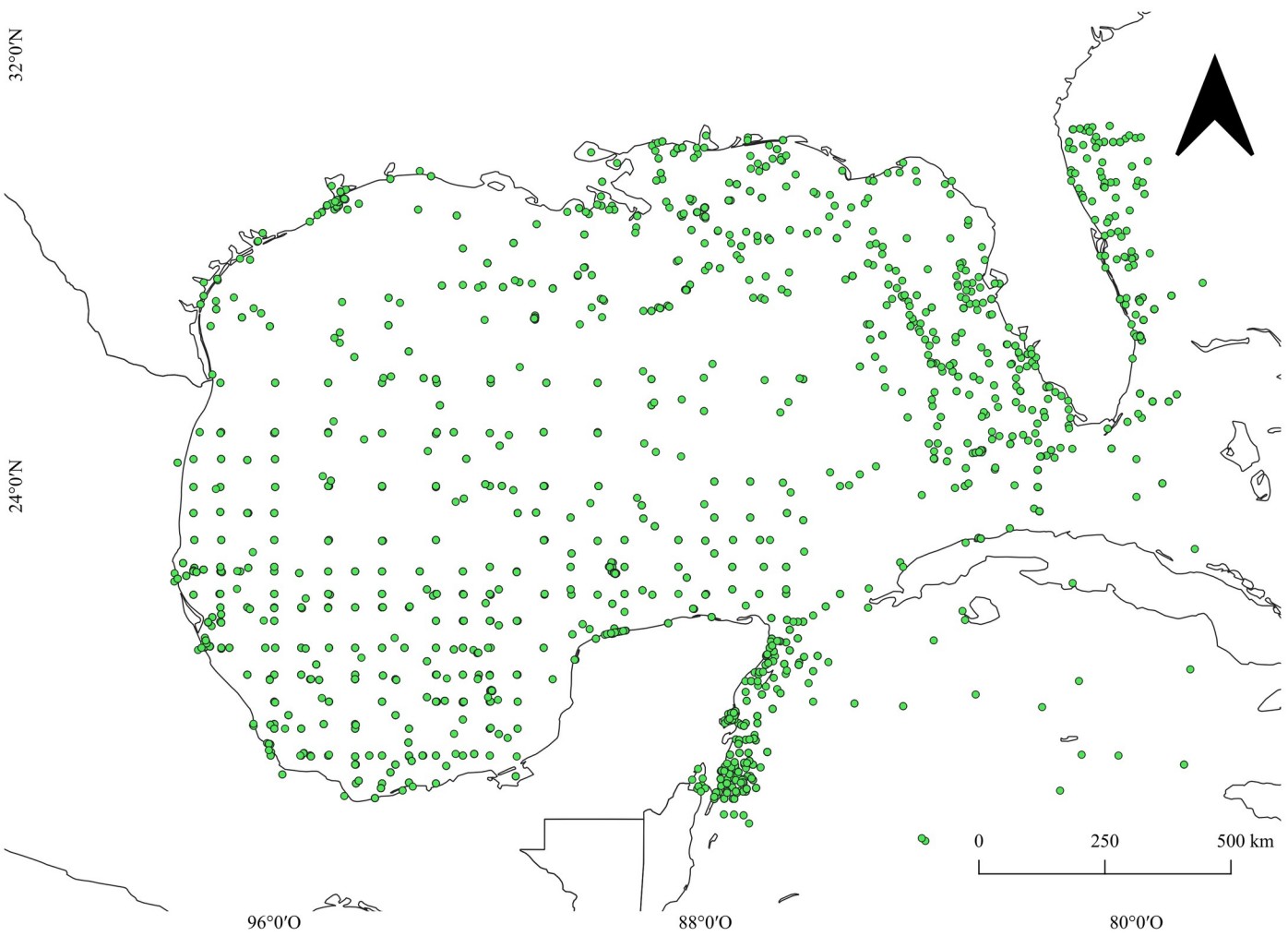

**Fig 1. Map of the study area with the occurrence records distribution (6067).**

[see 27]. Then, individual tracks supporting previously found clades were removed from the matrix, and a new run was performed. This process was repeated until no more generalized tracks were recognized. Successively nested tracks were considered as new generalized tracks to identify endopatric patterns. Each generalized track was graphed into a map by applying an MST algorithm to the occurrence records constituting the individual tracks that supported each clade. A panbiogeographical node was indicated in areas where two or more generalized tracks overlapped [see 27]. In addition, we considered the distribution of generalized tracks and nodes to recognize biogeographical areas, which can be useful in future attempts to regionalize the GoMx.

## Results

### Track analysis

After the first run, four cladograms were retained. The consensus cladogram had a length of 1893 steps, a consistency index of 0.123, and a retention index of 0.416. In this run, we recovered three generalized tracks (GT 1, GT 2, GT 3) and one nested track (NT 1) (Fig 2a andd

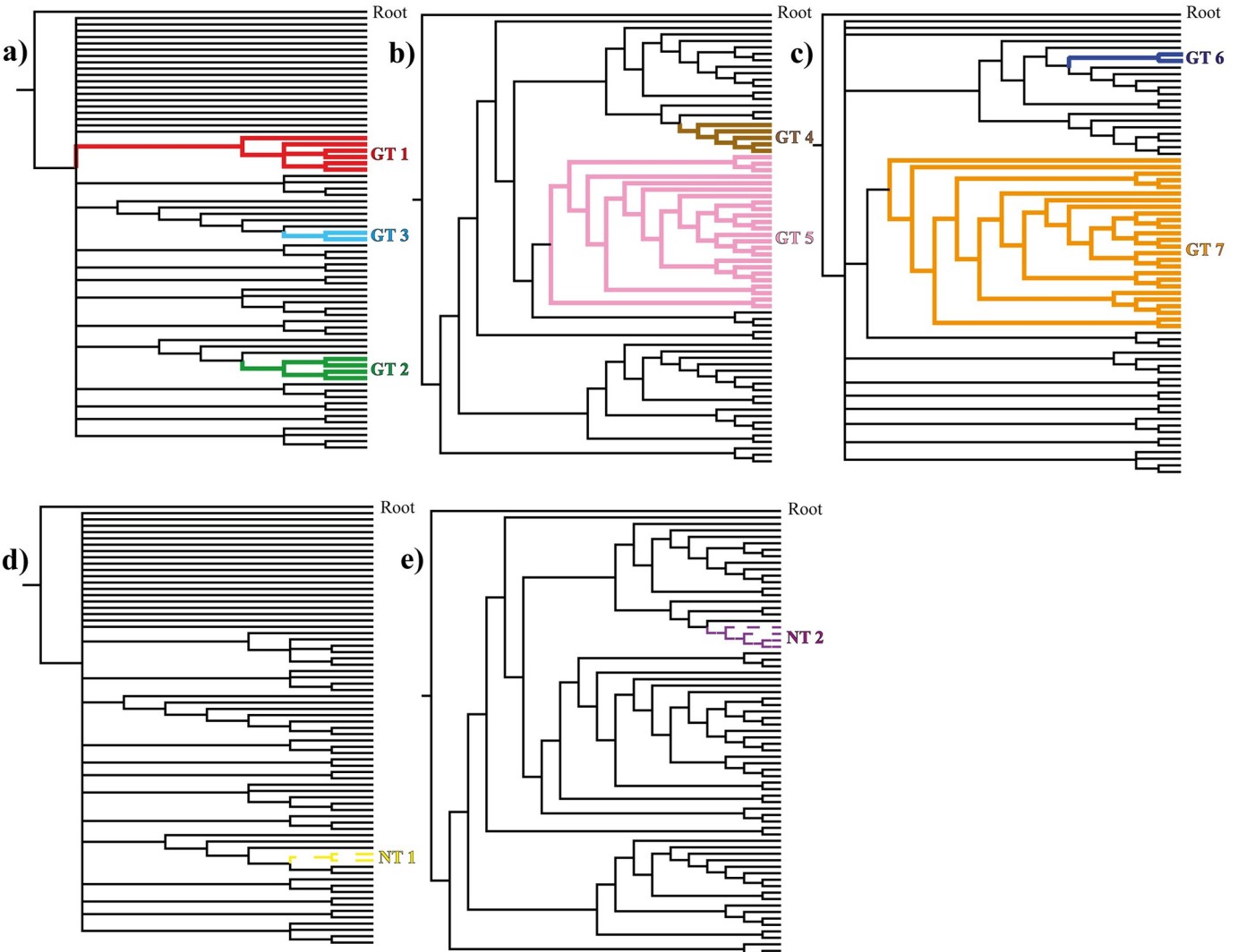

**Fig 2. Cladograms resulting from the parsimony analysis of endemicity with progressive character elimination (PAE-PCE), representing the three successive runs.** a)-c) Solid-colored branches represent clades supported by at least two synapomorphic individual tracks and are considered generalized tracks, respectively. d)-e) Dotted-colored branches represent nested tracks from the generalized tracks 2 and 4, respectively.

2d). GT 1 is located in the southern GoMx, in zones of continental shelf, off the states of Veracruz and Tabasco, along the Sistema Arrecifal Veracruzano (SAV). This generalized track is supported by the distributional congruence of five taxa (Fig 3a; Table 1). GT 2 is located in the northern GoMx, on the shelf of the Florida Peninsula, covering all of the Florida Keys Reef Tract, and is supported by three taxa (Fig 3a; Table 1), allowing the conformation of a nested pattern supported by four taxa (NT 1), located along the Dry Tortugas and Lower Keys (Fig 3a; Table 1). GT 3 is located in the Mexican Caribbean and is supported by two taxa (Fig 3a; Table 1).

The second run retained only one cladogram with 1408 steps, a consistency index of 0.156, and a retention index of 0.576. In this run, two generalized tracks (GT 4 and GT 5) and one nested pattern (NT 2) were recognized (Fig 2b and 2e). GT 4, located over the Florida Peninsula, is supported by two taxa. This generalized track contains a nested track (NT 2), supported

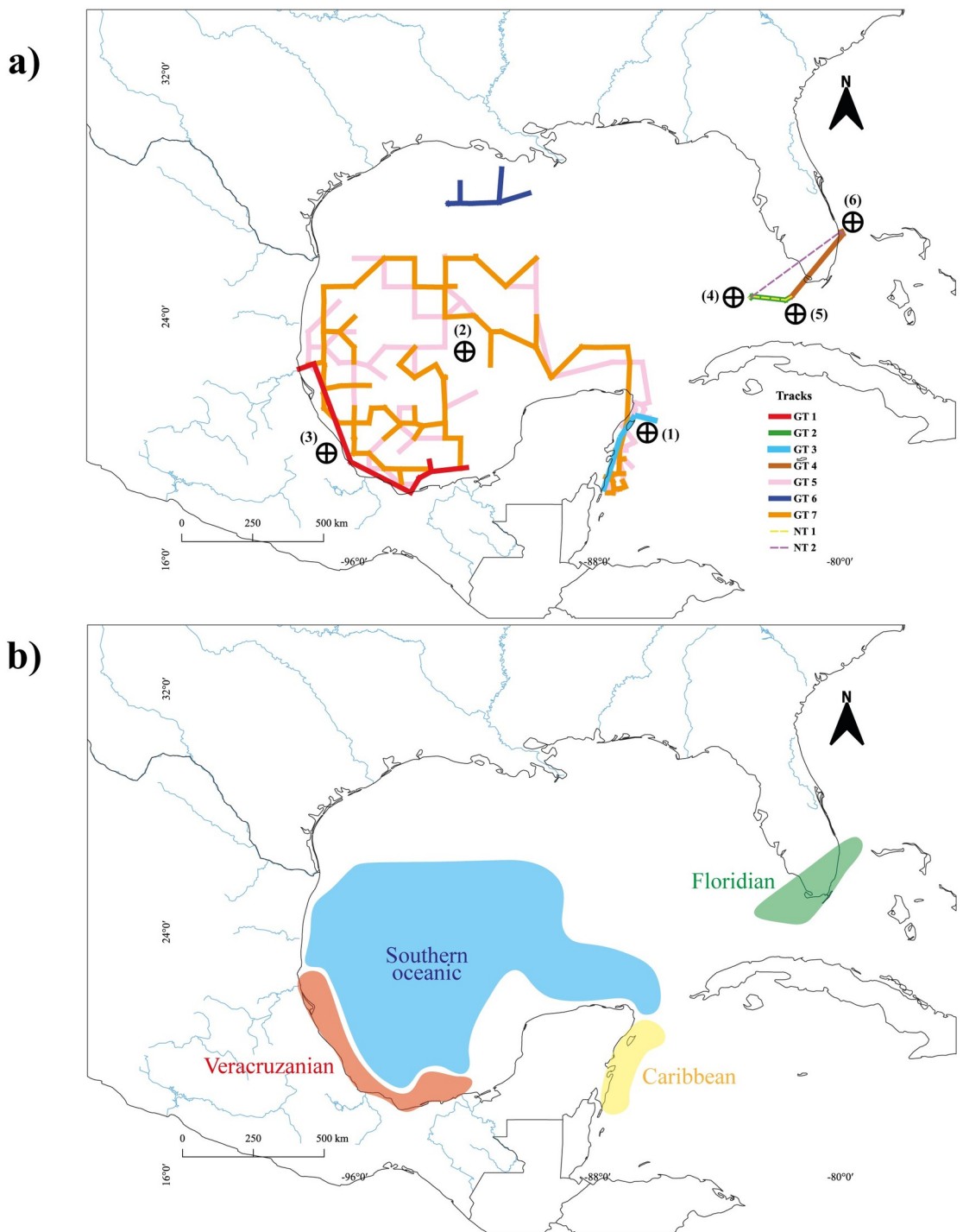

**Fig 3. Biotic assemblages of gelatinous zooplankton in the Gulf of Mexico and adjacent waters.** a) Generalized tracks (GT) indicated by solid-colored lines; nested tracks (NT) indicated by discontinuous colored lines; and panbiogeographical nodes indicated by incircle crosses. b) Biogeographical areas recognized by the distribution of generalized tracks and panbiogeographical nodes. Blue solid lines in the land represent the main rivers of the region.

**Table 1. Taxa supporting each generalized and nested tracks from the Gulf of Mexico and adjacent waters.**

| Run | Generalized tracks | Species | Taxa | Nested tracks | Species | Taxa |
|---|---|---|---|---|---|---|
| 1 | GT 1 | *Althoffia tumida* | Appendicularia | | | |
| | | *Appendicularia sicula* | Appendicularia | | | |
| | | *Fritillaria megachile* | Appendicularia | | | |
| | | *Oikopleura (Vexillaria) albicans* | Appendicularia | | | |
| | | *Tectillaria fertilis* | Appendicularia | | | |
| 1 | GT 2 | *Aequorea neocyanea* | Hydrozoa | NT 1 | *Aeginura grimaldii* | Hydrozoa |
| | | *Merga violacea* | Hydrozoa | | *Aequorea floridana* | Hydrozoa |
| | | *Orchistoma pileus* | Hydrozoa | | *Eutima coerulea* | Hydrozoa |
| | | | | | *Multioralis ovalis* | Hydrozoa |
| 1 | GT 3 | *Lensia achilles* | Hydrozoa | | | |
| | | *Thalia cicar* | Thaliacea | | | |
| 2 | GT 4 | *Staurodiscus kellneri* | Hydrozoa | NT 2 | *Forskalia edwardsii* | Hydrozoa |
| | | *Wuvula ochracea* | Hydrozoa | | *Proboscidactyla gemmifera* | Hydrozoa |
| | | | | | *Zancleopsis dichotoma* | Hydrozoa |
| 2 | GT 5 | *Lensia cossack* | Hydrozoa | | | |
| | | *Lensia meteori* | Hydrozoa | | | |
| 3 | GT 6 | *Lovenella grandis* | Hydrozoa | | | |
| | | *Neoturris pileata* | Hydrozoa | | | |
| 3 | GT 7 | *Dimophyes arctica* | Hydrozoa | | | |
| | | *Lensia hotspur* | Hydrozoa | | | |

by three taxa, going from the Dry Tortugas to the southwest of the peninsula (Fig 3a; Table 1). GT 5, supported by two taxa, covers a large area, passing through the Yucatán Channel in the Mexican Caribbean to a vast zone in the oceanic waters of the southwest GoMx (Fig 3a; Table 1).

In the third run, two cladograms were retained. The strict consensus cladogram of 1399 steps, a consistency index of 0.152, and a retention index of 0.568 allowed the identification of two generalized tracks (GT 6 and GT 7) (Fig 2c). In the fourth run, no more generalized tracks were identified. GT 6, supported by two taxa, is located between the Sigsbee scarp and the Mississippi Canyon in northern GoMx (Fig 3a; Table 1). Finally, GT 7 coincides spatially with GT 5, covering a large area from the Mexican Caribbean, surrounding the Yucatán Peninsula, and reticulating in the oceanic zones of the southwest GoMx. This generalized track is supported by two taxa (Fig 3a; Table 1).

## Nodes and complex areas

Six nodes are recognized and described: node (1) is located in the Yucatán Channel resulting from the overlapping of three generalized tracks; node (2), located in the oceanic region of southwest GoMx, is the result of the intersection of two generalized tracks, although in the neritic zone of this region, these tracks interact with GT 1; node (3) is recognized after the overlap of three generalized tracks along the coast of southwest GoMx, going from Veracruz to Tabasco. Finally, nodes (4), (5), and (6) are located in the northern GoMx, over the shelf of the Florida Peninsula, resulting from the overlap of two generalized tracks and two nested tracks (Fig 3a; Table 2).

We identified four biogeographic areas based on the distribution of the generalized tracks. The "Caribbean" is located in the Mexican Caribbean and characterizes the Yucatán Channel, a zone with important coral development and high hydrological dynamism. The "Caribbean"

**Table 2. Characterization of panbiogeographical nodes and biogeographic areas from the Gulf of Mexico and adjacent waters.**

| ID node | Overlapping generalized tracks | Hydrological, physical, geological, and biological features | Zones recognized as complex areas in this study | Biogeographic zones previously recognized |
|---|---|---|---|---|
| (1) | GT 3, GT 5, GT 7 | Yucatán current, coral reefs zone | Caribbean | Mesoamerican Caribbean platform [44], Neritic zone and slope of the Mesoamerican Caribbean [21] |
| (2) | GT 5, GT 7 | Oceanic mainly | Southern oceanic | Southern Gulf of Mexico slope, Gulf of Mexico basin (14.3, 14.3 in Wilkinson et al. [44]), Oceanic assemblage [14], Basin of the Gulf of Mexico [21] |
| (3) | GT 1, GT 5, GT 7 | Neritic, coral reefs zone, river influence | Veracruzanian | Veracruzan Neritic, Tabascan Neritic (14.1.1, 14.1.2 in Wilkinson et al. [44]), West coast of the Gulf of Mexico [21], Costa Occidental del Golfo de México [45] |
| (4) | GT 2, NT 1, NT 2 | Loop Current, coral reefs zone | Floridian | Floridian [24, 44] |
| (5) | GT 2, GT 4, NT 1 | Loop Current, coral reefs zone | Floridian | Floridian [24, 44] |
| (6) | GT 2 GT 4, NT 2 | Gulf current | Floridian | Floridian [24, 44] |

spatially interacts with another recognized area named "Southern Oceanic", which characterizes a vast oceanic zone in the southwest GoMx. The third area, "Veracruzanian", interacts with the oceanic zone of the southern GoMx but also characterizes a neritic region with large coral reefs. Finally, "Floridian" is an area that characterizes the southern Florida Peninsula in the northern GoMx, with a defined hydrological dynamism and diverse coral reefs (Fig 3b).

## Discussion

Different taxonomic groups of gelatinous zooplankton in the GoMx share distributional patterns, leading to the conformation of biotic assemblages, probably by a common biogeographical history, which allowed us to recognize a spatially structured distribution. This study contributes to the biogeographical baseline of the GoMx and aims to expand it in the future by incorporating time-slicing techniques, taxa dating, and biogeographic cladistic studies.

### Generalized tracks

The GoMx is an ecoregion where the biotas of tropical, subtropical, and temperate waters converge [25, 46], leading to a biotic complexity characterized by high biodiversity and endemism [14, 24], which is supported by the generalized tracks and nodes identified in this study. Most of the generalized tracks (GT 1, GT 2, GT 3, GT 4, NT 1, NT 2) and nodes (i.e., (1), (3), (4), (5), (6)) are located in neritic zones, which can be explained by the sharp physical and hydrological changes experienced in these zones throughout the different seasons of the year. These zones harbor biotas with restricted distributional ranges adapted to their characteristic environments (e.g., coastal lagoons, estuaries, bays, coral reefs, and rivers), which contribute greatly to endemism [47].

GT 1 and GT 3 coincide with generalized tracks previously identified with macroalgae and echinoderms in southern GoMx and the Mexican Caribbean [22, 45]. According to Vilchis et al. [22], the conformation of these biotas can be explained by two historical processes: 1) the emergence of the Isthmus of Tehuantepec in the early Eocene (~50 Ma ago), and 2) the emergence of the Yucatán Peninsula in the middle Miocene (~19 Ma ago). Here, we propose additional factors to explain these patterns, such as the Caribbean Current System, which favors the distributional convergence of several planktonic organisms by funneling water masses from the Atlantic Ocean into the GoMx, contributing to the conformation of biotic assemblages such as GT 3 [19, 48]. Likewise, the local current system of the southwest GoMx allows us to explain GT 1 [49] since high self-recruitment of some organisms has been reported in

this zone due to seasonal variability in the current direction over its narrow platform (i.e., southward during the fall-winter period and northward during the spring-summer period). These seasonal changes are associated with atmospheric conditions such as wind direction [19, 50] and influence the distribution of planktonic organisms that are current-dependent, contributing to the conformation of spatially restricted biotas.

GT 5 and GT 7 are supported by siphonophores, a group of hydrozoans with interesting ecological and biological characteristics. Siphonophores are colonial organisms with a modular body plan that commonly inhabit the open ocean [51]. These organisms represent an important component of gelatinous zooplankton populations in oceanic zones [52], although they can also be common inhabitants of neritic environments with a hard oceanic influence, such as the southwest GoMx [19]. This fact explains the wide distribution of these tracks, covering both oceanic and neritic zones. In addition, the oceanic region of the southern GoMx is physicochemically more homogeneous compared to neritic zones and has been proposed as a region biologically structured by nestedness [14], implying that this zone contains populations with a wide distribution range, to which others with successively more restricted distribution ranges are added [53]. Therefore, *Lensia cossack*, *L. meteori*, *L. hotspur*, and *Dimophyes arctica* (Table 1) conform to a biotic assemblage that characterizes the most general level of the biological structure of gelatinous zooplankton in the southwest oceanic region of the GoMx. Further studies involving other taxonomic groups and techniques are needed to improve our understanding of the biotic structure of this zone.

The nested patterns of GT 2 and GT 4 (NT 1 and NT 2, respectively) suggest a high biotic complexity resulting from the hydrological dynamism of the zone caused by the Loop Current [54]. In addition, the local current patterns on the inner shelf of the Florida Peninsula (southward during the fall-winter period and northward during the summer) may shape the distribution of gelatinous zooplankton at local scales [19]. These hydrological characteristics generate connectivity and self-recruitment between the Florida reefs and other regions of the GoMx, which characterize this zone of high biodiversity [19, 54]. Finally, GT 6 can be explained by the marine current pattern of the Flower Garden Banks (which are part of the National Marine Sanctuary Program) and the influence of the Mississippi River plume [55]. Regardless of the geographical distance, important larval connectivity between this zone and the southwest GoMx (Veracruz) has been recognized, compared to the northeast region (Florida) [49, 54], suggesting a biological corridor between both zones of the Gulf. These facts support the interesting biological characteristics and the complex history of the Flower Garden Banks [56], making their study and conservation crucial.

## Nodes and complex areas

Node (1) is located in the Mexican Caribbean, where diverse ecosystems have developed over history, such as coastal lagoons (e.g., Nichupté), bays (Chetumal, Espiritu Santo, and Ascención), and islands (Cozumel, Mujeres, and Contoy) [16]. In the geological history of the Mexican Caribbean, two events can explain the biotic complexity of this zone. The first is the counterclockwise rotation of the Yucatán block (~140–160 Ma) [18], a vicariant event that led to the opening of the GoMx and the conformation of its main marine currents, allowing the transport, convergence, settlement, and evolution of diverse taxa. The second event is the emergence of the Isthmus of Panama (~ 23–20 Ma and 8–6 Ma) [57, 58], which caused changes in the current patterns that led to speciation by fragmentation of populations or by mixing with other populations [59]. The Mexican Caribbean is characterized by a large extent of coral reefs, high hydrological dynamism, and biodiversity [16], being recognized in the ecoregions system proposed by Wilkinson et al. [44], which considers diverse natural

protected areas established in this region. Therefore, we suggest the Caribbean as a natural area within a biogeographic scheme (Fig 3b; Table 2); however, this hypothesis should be tested by conducting more studies with different taxonomic groups and techniques, such as cladistic biogeography.

Node (2) can be interpreted as the interaction of biotas transported from the Caribbean into the GoMx by the Loop Current, which remains confined to the southwest oceanic zone due to mesoscale oceanographic processes [17, 52, 60, 61]. The southern GoMx has been regionalized using different approaches, e.g., considering its ecological or physicochemical characteristics [62]; however, these proposals do not agree with the Southern Oceanic area proposed here and in other studies [e.g., 24, 62]. The idea that the oceanic southern GoMx is biologically more homogeneous than the northern region [14, 62] is supported in this study. Likewise, the repeated intersection of generalized tracks in this zone supports the hypothesis that this region constitutes a transitional zone between the Carolinian and Caribbean provinces [24].

Node (3) lies in a fracture zone along which the Yucatán/Chiapas massif and the central GoMx migrated southward [63]. This area is denoted by the Trans-Mexican Volcanic Belt and is characterized by volcanic coasts, numerous Pleistocene coral reefs, and river influence [64–66]. This zone originated from the tectonic activity of the Cocos and North American plates [67]. These characteristics support the recognition of this zone as a complex biogeographic area, distinct from the oceanic region of the southwest GoMx (Fig 3b). Previous studies have recognized the biotic complexity of this zone [22] or considered it within biogeographic schemes [44]. This area coincides with at least two protected natural areas, i.e., Sistema Arrecifal Lobos-Tuxpan and Sistema Arrecifal Veracruzano [68], thus supporting the hypothesis of recognizing the Veracruzanian as a natural area within a biogeographic scheme.

Nodes (4), (5), and (6) support the complex geological history of the Florida Peninsula, dating back to the Jurassic, when Pangea began to rift into the protocontinents of Laurasia and Gondwana. The peninsula continued its formation, becoming part of Gondwana [69]; however, successive rifting and collision events welded a fragment of the African plate to the North American plate, which explains the exotic geological characteristics of this zone, resembling Laurasia [70]. In addition, repeated events of suture, submergence, and emersion have shaped the distribution of some taxa in the northwest of Florida [57, 71]. The nodes of this zone are delimited by coral reefs that are ~ 125 ka old, constituting a zone with a long and complex geological history. This zone encompasses the entire Florida Keys Reef Tract, from Dry Tortugas National Park, Marquesas, Lower Keys, Middle Keys, and Upper Keys to Biscayne National Park (Fig 3a; Table 1). Therefore, it is considered a protected area due to its high biodiversity and complex ecosystems [54], with different zones considered as biogeographic areas [14, 24, 44] supporting its biotic complexity.

## Conclusions

The biotic assembly of gelatinous zooplankton in the GoMx can be explained by the dispersal-vicariance model [72]. In this sense, the main vicariant events of the region (e.g., the formation of the Isthmus of Panama and the rotation of the Yucatán Peninsula) allowed the convergence or isolation of diverse populations, leading to different speciation processes. Subsequently, the conformation of the main oceanic currents of the region favored biotic dispersal, shaping their current distribution through the local extinction process (due to competition or unfavorable environmental conditions) or by the colonization of new habitats. Moreover, coastal environments, such as lagoons or bays, represent the possibility of some populations being isolated

from the influence of marine currents, which favors the conformation of biotic assemblages with restricted distributional ranges.

The biogeography of the taxa from the GoMx is poorly understood compared to other regions and requires further research. Establishing a biogeographic baseline is fundamental to leading future multi-objective studies, allowing us to monitor changes in taxa distribution patterns over time due to current pressures (e.g., climate change [73, 74]). Zones of high biotic complexity in the GoMx spatially correspond to biogeographic areas of high biodiversity, supporting their naturalness and the utility of biogeography for identifying priority areas for conservation in marine environments.

Perspectives in the biogeography of the GoMx biotas include but are not limited to 1) increased sampling effort in oceanic regions, 2) accurate taxonomic identifications through integrative taxonomy, 3) cladistic studies to test proposed hypotheses, and 4) implementation of molecular clocks to shed light on divergence times of life and space. Knowing the biogeography of the GoMx biotas provides valuable information for future area biogeographic schemes, allowing us to improve our knowledge of the natural history of biological resources and generate better strategies for their management and conservation.

## Supporting information

**S1 File. Peer-reviewed literature consulted.**
(DOCX)

**S2 File. Occurrence data of gelatinous zooplankton analyzed in this study.**
(XLSX)

**S3 File. Community matrix implemented to conduct the PAE-PCE.**
(XLSX)

## Acknowledgments

This work is part of the doctoral project of J.M.A.-H. (CVU: 1079584).

## Author Contributions

**Conceptualization:** José María Ahuatzin-Hernández.

**Data curation:** José María Ahuatzin-Hernández.

**Formal analysis:** José María Ahuatzin-Hernández.

**Supervision:** Juan J. Morrone, Víctor M. Vidal-Martínez.

**Validation:** Juan J. Morrone, Víctor M. Vidal-Martínez.

**Writing – original draft:** José María Ahuatzin-Hernández, Juan J. Morrone, Víctor M. Vidal-Martínez.

**Writing – review & editing:** José María Ahuatzin-Hernández, Juan J. Morrone, Víctor M. Vidal-Martínez.

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
