## [Decision Letter · Decision Letter 0]

24 May 2024

PONE-D-23-39946Biotic assemblages of gelatinous zooplankton in the Gulf of Mexico and adjacent waters: An evolutionary biogeographic approachPLOS ONE

Dear Dr. Ahuatzin-Hernández,

Thank you for submitting your manuscript to PLOS ONE. After careful consideration, we feel that it has merit but does not fully meet PLOS ONE’s publication criteria as it currently stands. Therefore, we invite you to submit a revised version of the manuscript that addresses the points raised during the review process.

Let me start by apologizing for the time the review process took; As I already said, I had some difficulty finding reviewers;

Your manuscript was reviewed by two independent reviewers; Although both consider this to be a valid work and possibly worthy of publication, they also point out some flaws that I invite you to improve. Please clarify your goals (at this point they are not clear) and revise your discussion (needs some work) and clarify how the data used could have influenced the results

We look forward to receiving your revised manuscript.

Kind regards,

Clara F. Rodrigues

Academic Editor

PLOS ONE

4. "Thank you for stating the following financial disclosure: 

 [J.M.A.-H. received a scholarship from Consejo Nacional de Humanidades, Ciencia y Tecnología (CONAHCyT) for his doctoral research (grant: 845170) ].  

5. We note that [Figure(s) 1 and 3] in your submission contain [map/satellite] images which may be copyrighted. All PLOS content is published under the Creative Commons Attribution License (CC BY 4.0), which means that the manuscript, images, and Supporting Information files will be freely available online, and any third party is permitted to access, download, copy, distribute, and use these materials in any way, even commercially, with proper attribution. For these reasons, we cannot publish previously copyrighted maps or satellite images created using proprietary data, such as Google software (Google Maps, Street View, and Earth). For more information, see our copyright guidelines: http://journals.plos.org/plosone/s/licenses-and-copyright.

1. You may seek permission from the original copyright holder of Figure(s) [1 and 3] to publish the content specifically under the CC BY 4.0 license.  

Additional Editor Comments:

Reviewers' comments:

Reviewer's Responses to Questions

**Comments to the Author**

1. Is the manuscript technically sound, and do the data support the conclusions?

Reviewer #1: Partly

Reviewer #2: Partly

2. Has the statistical analysis been performed appropriately and rigorously? 

Reviewer #1: N/A

Reviewer #2: Yes

3. Have the authors made all data underlying the findings in their manuscript fully available?

Reviewer #1: Yes

Reviewer #2: Yes

4. Is the manuscript presented in an intelligible fashion and written in standard English?

Reviewer #1: Yes

Reviewer #2: No

5. Review Comments to the Author

Reviewer #1: Dear editor and authors,

This manuscript provides a comprehensive study of the biogeographical distribution and ecological dynamics of gelatinous zooplankton in the Gulf of Mexico (GoMx) and Western Caribbean. The study focuses on several taxa of gelatinous zooplankton, including Hydrozoa, Scyphozoa, Cubozoa, Ctenophora, Chaetognatha, Thaliacea, and Appendicularia, highlighting their evolutionary significance and ecological roles within marine ecosystems. The authors employ a panbiogeographical analysis, utilizing data from the Global Biodiversity Information Facility (GBIF) and the Ocean Biodiversity Information System (OBIS), supplemented with peer-reviewed literature. They apply track analysis, specifically a parsimony analysis of endemicity with progressive character elimination (PAE-PCE), to reveal spatial structures and biotic components. The findings suggest distinct biogeographic patterns and potential regions of endemism influenced by both historical and current oceanographic conditions, significantly contributing to our understanding of marine biodiversity and its conservation in the region. The study's approach and analytical methods are well-suited to address the complex interactions and evolutionary history of these taxa, providing a valuable framework for future biogeographical and ecological research in marine environments.

Some issues that could be better addressed or that still represent limitations of the study:

• Can we rely on generalized tracks composed of few species? Can we have some indication as to the robustness of the results or are all tracks equally reliable?

• In such a broad dataset, aren't these results from the generalized tracks relatively incipient, given that most tracks are supported by only two species?

• It is not clear how the authors relate the temporal component within the analyses. How do they associate spatial patterns with temporal patterns without having any explicit data on species divergence time?

• To what extent are panbiogeographic nodes representations of spatial biogeographic patterns or merely a result of sampling effort?

• The authors claim that the area is composed of complex biodiversity, which can be confirmed by the seven generalized tracks found. What constitutes a complex area? How many tracks form a complex area?

• How do the generalized tracks represent areas of interest for conservation? Aren't there more appropriate methods to deal with conservation-related approaches?

• How can we ensure that species showing shared distribution patterns are the result of the same historical processes without including time in the analyses?

• The authors state that the results can be explained by a dispersal-vicariance model. What could not be explained by this model?

The work certainly presents novel results for the area and the groups studied, although they are incipient regarding the amount of evidence supporting the patterns found. A greater number of species supporting the generalized tracks would be desirable, or additional analyses that could reveal other patterns that track analyses fail to uncover. I believe that if new data are added to the analyses or if other methods are applied to the same dataset, new results may emerge. I think it is important for the authors to address the questions I have raised, which could enhance the impact and reach of the work.

Reviewer #2: General comments

This study uses data from publicly available databases and published literature to conduct a track analysis to identify distributional patters of gelatinous zooplankton in the Gulf of Mexico.

From reading the paper, it is not exactly clear what the objectives of the study are. This needs to be much clearer from the start. Right now it reads like a bunch of already published data was gathered and then some analysis was conducted without a clear sense of direction.

The data sources were not described in any detail, which I found a bit odd, as it has the potential to drive the result of the analysis.

The conclusions were a bit too generic for my liking, saying more studies are needed shouldn’t really be a conclusion, suggest those next steps, preferably with clear objectives for future research.

Specific comments

Abstract

Line 26-27: Brief description of these databases would be helpful

Line 27: First time tracks are mentioned, some brief description of the analysis would be helpful.

Lines 36-37. Pretty sure some of the biotas are quite well studied, so be more specific there. Also missing why this kind of work is important.

Lines 38-39: Weak statement, and unclear how any of this relates to management at this point.

Introduction

As stated above objectives are sorely missing

Line 53: Stating that something is interesting is not really helpful, explain why it is interesting.

Line 70: Further description of this current would be helpful

Lines 72-74: Confusing sentence, I would turn it around and state that the features are likely to drive spatial structuring.

Lines 84-87. This needs to be expended, track analysis are (as stated) rarely used in the marine environment so readers from that realm need extensive introduction to the analysis and its uses.

Material and methods

As stated above, it would be beneficial to describe the data used in much more detail. Year range, depth range, methods of collection etc.

Results

Line 140. This figure legend is not very descriptive and does not stand alone. Describe better what the figure is showing.

Discussion.

As already mentioned, the influence of the data sources on the results are not mentioned at all. This needs to be addressed.

Conclusions:

Line 285-287. Confusing sentence, would rephrase.

6. PLOS authors have the option to publish the peer review history of their article (what does this mean?). If published, this will include your full peer review and any attached files.

Reviewer #1: No

Reviewer #2: No

---

## [Author Response · Author response to Decision Letter 0]

5 Jul 2024

Dear Professor Clara F. Rodrigues, Academic Editor and Reviewers,

Thank you in advance for the editorial handling and comments provided on our manuscript “Biotic assemblages of gelatinous zooplankton in the Gulf of Mexico and adjacent waters: An evolutionary biogeographic approach”. As indicated, we have followed all the comments/suggestions by the reviewers. In general, all the points and suggestions were addressed, considered, and integrated into the new version of our manuscript. The objective of the study is clearly stated in the introduction. The discussion section was improved considering the comments and the reach of our analysis. Conclusions are expanded and set in the jargon of the methodology and results. The color of GT 7 in Figures 2 and 3 was changed. We also revised the English of the document, providing a clearer and more fluid version. A point-by-point response to each comment is provided below. 

Editorial Requirements

4. "Thank you for stating the following financial disclosure: 

 [J.M.A.-H. received a scholarship from Consejo Nacional de Humanidades, Ciencia y Tecnología (CONAHCyT) for his doctoral research (grant: 845170) ]. 

R = The statement was added in the new version of the manuscript, as indicated (lines 319-320)

5. We note that [Figure(s) 1 and 3] in your submission contain [map/satellite] images which may be copyrighted. All PLOS content is published under the Creative Commons Attribution License (CC BY 4.0), which means that the manuscript, images, and Supporting Information files will be freely available online, and any third party is permitted to access, download, copy, distribute, and use these materials in any way, even commercially, with proper attribution. For these reasons, we cannot publish previously copyrighted maps or satellite images created using proprietary data, such as Google software (Google Maps, Street View, and Earth). For more information, see our copyright guidelines: http://journals.plos.org/plosone/s/licenses-and-copyright.

R = The figures presented in the manuscript do not constitute satellite images (either from Google maps, Street View or Earth). These figures were built with free download vectorial archives. In particular, the used vector can be downloaded from http://tapiquen-sig.jimdo.com. This vector is based on the shapes of the Environmental Systems Research Institute (ESRI), which is of free distribution. Therefore, no permission is required regarding the publication/distribution of figures 1, 3.

Reviewer # 1

Some issues that could be better addressed or that still represent limitations of the study:

• Can we rely on generalized tracks composed of few species? Can we have some indication as to the robustness of the results or are all tracks equally reliable?

R = An advantage of identifying generalized tracks using parsimony analysis of endemicity (PAE) is that generalized tracks represent a homology of areas. Because PAE is analogous to traditional systematic analyses, in which natural/monophyletic groups are recognized by synapomorphies, in our study, generalized tracks are natural areas constituting biogeographical hypotheses of area relationships. In this sense, the number of synapomorphies is not that relevant to validate the analysis, but the number of units (quadrants) integrated into a “phylogeny of areas” supported by at least two synapomorphic individual tracks. An indirect indication of the robustness of the results is provided by comparing the patterns obtained in our study with those of other studies that recovered spatially concordant generalized tracks with other taxonomic groups. If different taxa with different dispersal methods (benthonic or planktonic) share distributional patterns in a delimited geographic area, a biogeographical hypothesis of a common biogeographic history can be raised, likely due to dispersal and vicariant events. Some examples of studies recovering similar patterns in the GoMx are in:

• Vilchis MI, Dreckmann KM, García-Trejo EA, Hernández OE, Sentíes A. Patrones de distribución de las grandes macroalgas en el golfo de México y el Caribe mexicano: Una contribución a la biología de la conservación. Rev Mex Biodivers. 2018; 89: 183–192.

• Caballero-Ochoa AA, Martínez-Melo A, Conejeros-Vargas CA, Solís-Marín FA, Laguarda-Figueras A. Diversidad, patrones de distribución y “hotspots” de los equinoideos irregulares (Echinoidea: Irregularia) de México. Rev Biol Trop. 2017; 65: S42−S59.

• In such a broad dataset, aren't these results from the generalized tracks relatively incipient, given that most tracks are supported by only two species?

R = As stated above, while a clade is supported by at least two synapomorphic individual tracks, a hypothesis of an area relationship can be made. In addition, we have only three generalized tracks supported by two species. Consider the nested patterns; although at the general level only two species support the clade, nested endemism, also supported by at least two species, exists in two generalized tracks. Species supporting nested patterns also contribute to the general interpretation of the patterns. 

• It is not clear how the authors relate the temporal component within the analyses. How do they associate spatial patterns with temporal patterns without having any explicit data on species divergence time?

R = We do not intend to integrate the temporal component in our study. The study only aims to identify distributional patterns from a historical/evolutionary perspective. 

• To what extent are panbiogeographic nodes representations of spatial biogeographic patterns or merely a result of sampling effort?

R= Generalized tracks are the result of strict concordant and exclusive individual distributions. In addition, Fig 1 presents the distribution of records, showing a general homogeneous distribution throughout the study area, except for the Mexican Caribbean. For example, we did not recover tracks or nodes on either side of the Florida peninsula, which has an apparently major density of records. This occurs because none of the species distributed in these zones shared congruent distributions. In contrast, we found a track in the northern Gulf, in a zone with fewer records than the previously mentioned. Also, a node in the oceanic region of the Gulf was identified, a zone with fewer records than the continental shelf. These facts allowed us to conclude that even when sampling effort could influence the data analysis, PAE is a good tool to address this issue, uncovering patterns in zones with different densities of records, considering exclusively shared presences, and avoiding aspects that could bias the result, such as absents, abundances or densities. 

• The authors claim that the area is composed of complex biodiversity, which can be confirmed by the seven generalized tracks found. What constitutes a complex area? How many tracks form a complex area?

R = In panbiogeography, a complex area is identified by the overlap of two or more generalized tracks (Escalante et al. 2017). Because generalized tracks conceptually represent ancestral biotas, their interaction deserves a historical explanation. A brief explanation of this concept was added in lines 85-100 of the clean new version.

• Escalante, T., Noguera-Urbano, E. A., Pimentel, B., & Aguado-Bautista, O. (2017). Methodological issues in modern track analysis. Evolutionary Biology, 44, 284-293.

• How do the generalized tracks represent areas of interest for conservation? Aren't there more appropriate methods to deal with conservation-related approaches?

R = This is not the main objective of track analysis, although it has been implemented for conservational studies. This tool allows us to consider conservation perspectives because 1) it allows us to identify areas based on the restricted congruence of two or more species (i.e., the diagnostic species don’t occur in any other zone of the study area), 2) since the areas are recognized based on endemism, we can diagnose those areas in terms of their biodiversity, endemism, geographical boundaries, and abiotic characteristics. The main objective of our study was not to conduct a conservation biogeography analysis, but we considered it important to recognize that our areas match priority zones for conservation previously recognized. An example of a study implementing track analysis for conservation biogeography is provided below. 

• Luna-Vega I, Morrone JJ, Escalante T. Conservation biogeography: A viewpoint from evolutionary biogeography. In: Gailis M, Kalnin S, editors. Biogeography. New York, Nova-Science Publishers; 2010. pp 229–240.

• How can we ensure that species showing shared distribution patterns are the result of the same historical processes without including time in the analyses?

R = This is a concept of panbiogeographical studies. If a distribution pattern repeats through different taxa, a biota can be recognized, and the space delimited by the coincident distributions becomes significant for a historical interpretation. The idea of a shared biogeographic history is raised as a hypothesis, which can be tested with further studies. See lines 88-94 of the new clean manuscript. 

• The authors state that the results can be explained by a dispersal-vicariance model. What could not be explained by this model?

R = Historically, biogeographical explanations of species distribution were attributed either to dispersal or vicariance. According to the dispersal-vicariance model, both processes are significant to understanding the current species distribution. As these concepts are currently discussed in the literature, we consider important mentioning them. Some examples arguing the concepts are provided below:

Crisci, J. V., & Katinas, L. (2009). Darwin, historical biogeography, and the importance of overcoming binary opposites. Journal of Biogeography, 36(6), 1027-1032.

Morrone, J. J. (2020). Biotic assembly in evolutionary biogeography: a case for integrative pluralism. Frontiers of Biogeography, 12, e48819.

Reviewer # 2

From reading the paper, it is not exactly clear what the objectives of the study are. This needs to be much clearer from the start. Right now it reads like a bunch of already published data was gathered and then some analysis was conducted without a clear sense of direction.

R = The objective of the study was clarified. See lines 97-100

The data sources were not described in any detail, which I found a bit odd, as it has the potential to drive the result of the analysis.

R = Data sources are described in the new version regarding criteria implemented to gather the information. See lines 103-111

The conclusions were a bit too generic for my liking, saying more studies are needed shouldn’t really be a conclusion, suggest those next steps, preferably with clear objectives for future research.

R = This section was improved by adding some perspectives on future biogeographic studies of the region (lines 306-313)

Specific comments

Abstract

Line 26-27: Brief description of these databases would be helpful

R = A brief description of the dataset (regarding the number of occurrences) was added in the abstract. More specific details on this point were described in the Material and Methods section (lines103-111)

Line 27: First time tracks are mentioned, some brief description of the analysis would be helpful.

R = A description of the track analysis was added in the final paragraph of the Introduction section

Lines 36-37. Pretty sure some of the biotas are quite well studied, so be more specific there. Also missing why this kind of work is important.

R = These points were addressed. The sentences were changed as follows: “The historical biogeography of the gelatinous biotas from the Gulf of Mexico has been little studied compared with ecological approaches….” “This type of research is fundamental to understanding the evolutionary history of natural resources from a spatial perspective….” (lines 37-41)

Lines 38-39: Weak statement, and unclear how any of this relates to management at this point.

R = This statement was modified (lines 38-41)

Introduction

As stated above objectives are sorely missing

R = The objectives are clearly declared in the new clean version (lines 97-100)

Line 53: Stating that something is interesting is not really helpful, explain why it is interesting.

R = This sentence was changed as follows: “Hence, the study of gelatinous zooplankton as a functional group is promising for recognizing convergent evolutionary patterns regarding spatial dynamics and biogeography” (lines 53-55)

Line 70: Further description of this current would be helpful

R = A description of the marine current system of the Gulf of Mexico and adjacent water was considered (lines 70-76)

Lines 72-74: Confusing sentence, I would turn it around and state that the features are likely to drive spatial structuring.

R = This sentence was modified as suggested, i.e., “Hence, these features are likely to drive spatial structuring in the GoMx’s biotas, resulting from historical and oceanographic events” (lines 75-76)

Lines 84-87. This needs to be expended, track analysis are (as stated) rarely used in the marine environment so readers from that realm need extensive introduction to the analysis and its uses.

R = The explanation of track analysis was expanded, as suggested (lines 85-100)

Material and methods

As stated above, it would be beneficial to describe the data used in much more detail. Year range, depth range, methods of collection etc.

R = We did not consider year range, depth range, or sampling method as criteria since many records of gelatinous zooplankton from OBIS or GBIF lack this information. The criteria implemented to gather the information are explained in the Data acquisition and processing section (lines 105-108) 

Results

Line 140. This figure legend is not very descriptive and does not stand alone. Describe better what the figure is showing.

R = The figure legend was re-described, as suggested (lines 150-154)

Discussion.

As already mentioned, the influence of the data sources on the results are not mentioned at all. This needs to be addressed.

R = The characteristics of the data source (s) are expanded in the Data acquisition and processing section. We decided to consider only records with taxonomic and geographical certainty by querying information from scientific collections (e.g., museums) and published literature (lines 103-114).

Conclusions:

Line 285-287. Confusing sentence, would rephrase.

R = The Conclusion section was rephrased, adding new perspectives to the study of biogeography in the Gulf of Mexico. This particular sentence was changed as follows: “Zones with major biotic complexity spatially match biogeographic areas with high biodiversity, supporting their naturalness and demonstrating the utility of biogeography for recognizing priority sites for conservation in marine environments” (lines 307-310)

---

## [Editor Report · Decision Letter 1]

16 Jul 2024

Biotic assemblages of gelatinous zooplankton in the Gulf of Mexico and adjacent waters: An evolutionary biogeographic approach

PONE-D-23-39946R1

Dear Dr. Ahuatzin-Hernández,

We’re pleased to inform you that your manuscript has been judged scientifically suitable for publication and will be formally accepted for publication once it meets all outstanding technical requirements.

Kind regards,

Clara F. Rodrigues

Academic Editor

PLOS ONE

Additional Editor Comments (optional):

Thank you for addressing the reviewers' questions.
---

## [Editor Report · Acceptance letter]

19 Jul 2024

PONE-D-23-39946R1 

PLOS ONE

Dear Dr. Ahuatzin-Hernández, 

I'm pleased to inform you that your manuscript has been deemed suitable for publication in PLOS ONE. Congratulations! Your manuscript is now being handed over to our production team.

Kind regards, 

on behalf of

Dr. Clara F. Rodrigues 

Academic Editor

PLOS ONE